# Management of Liver Tumors during the COVID-19 Pandemic: The Added Value of Selective Internal Radiation Therapy (SIRT)

**DOI:** 10.3390/jcm10194315

**Published:** 2021-09-22

**Authors:** Irene Bargellini, Giuseppe Boni, Antonio Claudio Traino, Elena Bozzi, Giulia Lorenzoni, Francesca Bianchi, Rosa Cervelli, Tommaso Depalo, Laura Crocetti, Duccio Volterrani, Roberto Cioni

**Affiliations:** 1Department of Interventional Radiology, Pisa University Hospital, 56126 Pisa, Italy; elenabozzi@libero.it (E.B.); lorenzoni.giulia@hotmail.it (G.L.); rosacervelli.med@gmail.com (R.C.); laura.crocetti@med.unipi.it (L.C.); r.cioni@ao-pisa.toscana.it (R.C.); 2Department of Nuclear Medicine, Pisa University Hospital, 56126 Pisa, Italy; bonigb@yahoo.it (G.B.); fbianchimednuc@yahoo.it (F.B.); tom.depalo@gmail.com (T.D.); duccio.volterrani@unipi.it (D.V.); 3Department of Health Physics, Pisa University Hospital, 56126 Pisa, Italy; c.traino@ao-pisa.toscana.it

**Keywords:** radioembolization, yttrium, holmium, hepatocellular carcinoma, cholangiocarcinoma, liver cancer, metastases, COVID-19, pandemic

## Abstract

Background: In the context of the coronavirus disease 2019 (COVID-19) pandemic, liver-directed therapies (LDTs) may offer minimally invasive integrative tools for tumor control. Among them, selective internal radiation therapy (SIRT) represents a safe, flexible and effective treatment. Purpose of this study is to present our experience with SIRT during the first wave of COVID-19 pandemic and provide an overview of the indications and challenges of SIRT in this scenario. Methods: We retrospectively analyzed the number of patients evaluated by Multidisciplinary Liver Tumor Board (MLTB) and who were undergoing LDTs between March and July 2020 and compared it with 2019. For patients treated with SIRT, clinical data, treatment details and the best radiological response were collected. Results: Compared to 2019, we observed a 27.5% reduction in the number of patients referred to MLTB and a 28.3% decrease in percutaneous ablations; transarterial chemoembolizations were stable, while SIRT increased by 64%. The majority of SIRT patients (75%) had primary tumors, mostly HCC. The best objective response and disease control rates were 56.7% and 72.2%, respectively. Conclusion: The first wave of the COVID-19 pandemic was characterized by an increased demand for SIRT, which represents a safe, flexible and effective treatment, whose manageability will further improve by simplifying the treatment workflow, developing user-friendly and reliable tools for personalized dosimetry and improving interdisciplinary communication.

## 1. Introduction

The worldwide occurrence of the coronavirus disease 2019 (COVID-19) pandemic is having a profound effect on the management of cancer patients. Delays have been reported in referrals, screenings, diagnoses, surgical programs and follow-ups, as well as modifications of systemic treatments’ regimens, suspensions and difficulties in accessing clinical trials and progressive implementation of treatments allowing for self-care management and remote consultations [1,2,3,4].

Over the past years, liver-directed therapies (LDTs), such as percutaneous ablation and transarterial treatments, have received wide recognition in international guidelines, both for primary and metastatic liver lesions [5,6,7,8]. Low invasiveness, safety and good local tumor control, limited resource consumption and possible combination with other local and systemic therapies represent major advantages of LDTs. In the context of the COVID-19 pandemic, LDTs have become an essential tool for the management of cancer patients. Among LDTs, intra-arterial selective internal radiation therapy (SIRT) represents a safe, well-tolerated and effective treatment option in well-selected, non-surgical patients with radiosensitive liver tumors, provided adequate tumor targeting and that a sufficient radiation dose is delivered to the tumor [9,10,11,12,13,14].

Purpose of this study is to analyze the utilization of SIRT in liver cancer patients during the first wave of the COVID-19 pandemic, and to provide an overview of the indications and the challenges of SIRT in this new scenario.

## 2. Materials and Methods

In Italy, the COVID-19 outbreak forced a first lockdown from March to May 2020, allowing only patients in emergency situations access to care as well as oncologic patients at high risk of early progression, defined as those requiring treatment within 30 days.

In our tertiary referral university hospital, this period was characterized by a sudden block in patients’ referrals, reduced access to any procedure requiring anesthesiologic support, lack of blood donations and forced annulment of surgical and transplantation procedures.

Despite this situation, selected oncologic activities were maintained, including some loco-regional therapies not requiring anesthesiologic support, blood transfusions and longer hospitalization.

We retrospectively reviewed the registries of our weekly Multidisciplinary Liver Tumor Board (MLTB) to assess the number of patients who were evaluated from March (when the lockdown started) until July 2020 (to include the phase of exiting the lockdown) and compared it with the same period of time in 2019. Our electronic database was searched to assess the number of LDTs performed after MLTB discussion in the same period of time. LDTs included percutaneous ablation, transarterial chemoembolization (TACE) and SIRT.

For patients treated with SIRT, demographic and clinical data were collected, including tumor histology and stage, indications for SIRT and treatment modality.

### 2.1. SIRT Workflow

SIRT was preceded by a simulation procedure consisting of an angiographic study, with transfemoral arterial access, to detect arterial feeders supplying the lesions and any extrahepatic branches requiring preventive embolization.

After the catheter was placed in its final position, Technetium-99m macroaggregated albumin (Tc-99m MAA) or scout dose Holmium-166-labeled particles were injected intra-arterially, and SPECT-CT was performed to assess hepato-pulmonary shunting, extra-hepatic depositions and target lesion uptake and to perform dosimetric calculations.

The treatment was performed 1–3 weeks after the preliminary evaluation, placing the microcatheter in the planned position and administering the radiolabeled microspheres (SirSphere^®^, Sirtex Medical Europe GmbH, Bonn, Germany; TheraSphere^®^, Boston Scientific, Marlborough, MA, USA; QuiremSpheres^®^, Terumo Europe NV, Leuven, Belgium).

In case of bilobar disease, the treatment of each lobe was performed in separate sessions, at an interval of 4–6 weeks.

### 2.2. Follow-Up

Patients were discharge 24–48 h after the procedure. Clinical and radiological follow-up was performed at 45 days and every 3 months thereafter. Imaging follow-up was performed by CT and/or MR.

For the present study, the best radiological tumor response was collected only for patients treated in 2020, and it was assessed according to the Response Evaluation Criteria for Solid Tumors (RECIST) 1.1 [15] for metastatic lesions and intrahepatic cholangiocarcinoma (iCC) and modified RECIST (mRECIST) [16] for hepatocellular carcinoma.

### 2.3. Statistical Analysis

Data were analyzed using descriptive statistics (mean and standard deviation, SD) and compared with the Chi-square or Fisher’s exact test for categorical data and Student’s *t*-test for paired data. Statistical analysis was carried out with dedicated software (SAS, Cary, NC, USA) considering a *p* value < 0.05 as statistically significant.

## 3. Results

Between March and July 2020, 353 cases were discussed at the MLTB, compared to 487 cases of 2019, resulting in 27.5% reduction; this decrease was more evident in April and May with a slow resumption after May 2020 (Figure 1a).

During the same period of time, the number of percutaneous ablations decreased by 28.3% (from 60 procedures in 2019 to 43 procedures in 2020, Figure 1b), while TACE remained stable (63 procedures in 2019 and 64 in 2020, Figure 1c). On the other hand, the number of SIRT treatments increased by 64%, from 25 procedures in 22 patients in 2019 to 41 procedures in 36 patients in 2020 (Figure 1d).

### 3.1. Patients’ Characteristics

Overall, 31 (75.6%) procedures were performed in primary lesions in 2020 (mostly HCC), compared to 14 procedures (56%) in 2019.

Main demographic and clinical data of patients treated with SIRT are reported in Table 1.

According to the Barcelona Clinic of Liver Cancer (BCLC) staging system [5], over 50% of HCC patients were in the intermediate stage (BCLC B), while approximately one-third of cases were in the advanced stage (BCLC C) due to intrahepatic macrovascular invasion (Table 1). No early-stage HCC patients underwent SIRT between March and July 2020, compared to three (25%) cases treated in 2019 (*p* = 0.04).

SIRT in iCC was performed as first-line treatment in two cases, and after first-line chemotherapy in the remaining cases, either as consolidation after radiological disease control (*n* = 3) or at progression (*n* = 2).

Metastatic lesions included mostly patients with liver-only or liver-dominant colorectal cancer metastases (mCRC). In 2019, two cases with neuroendocrine and medullary thyroid carcinoma metastases, respectively, underwent SIRT as first-line treatment modality, whereas all the other cases were treated after failure of second- or third-line systemic chemotherapy.

### 3.2. Treatment 

Treatment modalities did not differ significantly comparing 2019 and 2020 (Table 1); the majority of procedures were performed as lobar procedures, with approximately 20% of cases requiring bilobar treatment. Patients were mostly treated with Yttrium-90 (Y-90)-labeled resin microspheres, although in 2020, the number of procedures performed using Holmium-166-labeled microspheres increased (19.5% in 2020 compared to 9.1% in 2019).

### 3.3. Tumor Response

No complications were observed after SIRT, and patients were discharged 24–48 h after treatment, according to our institutional policies.

Radiological outcomes were evaluated only for patients treated in 2020 and were available in 30/36 (83.3%) patients. Considering the best radiological response, the objective response rate was 56.7% (17/30) and the disease control rate was 72.2%, with four patients showing progressive disease early after treatment (Table 2). The best radiological response was registered 1–6 months after treatment (mean ± SD, 3.07 ± 1.46 months).

## 4. Discussion

During the first wave of COVID-19 pandemic, oncologic patients suffered from a sudden limitation in surgical procedures and percutaneous ablations, reduced access to more demanding systemic therapies requiring hospitalization or close monitoring and limited availability of clinical trials [1,2,3,4]. In our experience, in this context, despite the overall reduction in the number of referrals and in the number of LDTs requiring anesthesiological support (such as percutaneous ablation), SIRT increased by 64% compared to the same interval of time of the previous year.

SIRT is safe and well-tolerated, even in more fragile and elderly patients [17]; no complications were observed in our series and all patients were discharged within 48 h after the procedure, according to our hospital policy. Moreover, SIRT is highly flexible; it can be indicated in a number of primary and secondary radiosensitive liver tumors [9,10,11,12,13,14], and it does not preclude other concomitant or subsequent systemic or surgical treatments. The latter represented a highly appealing feature during the COVID-19 breakthrough, allowing doctors to control tumor progression or even reduce the tumor load while patients were waiting for surgical evaluation or access to other therapies.

### 4.1. Indications for SIRT

The majority of the procedures performed in 2020 involved patients with primary liver lesions (75.6% compared to 56% in 2019), with most of all HCC in the intermediate and advanced stages.

#### 4.1.1. Hepatocellular Carcinoma

According to the national recommendations, during the lockdown, priority was given to oncologic patients at risk of rapid progression. Treatments of early-stage HCC patients were thus often postponed.

Meanwhile, clinical trials investigating new systemic drugs in intermediate/advanced stage HCC were temporarily suspended. Considering the toxicity of the standard-of-care systemic therapies and the difficulties in monitoring patients under treatment, SIRT represented a valid option in BCLC C patients with liver-limited disease (such as patients with intrahepatic macrovascular invasion) and in BCLC B patients with a tumor extension considered unfit for TACE.

In this setting, large prospective randomized studies have failed in demonstrating the superiority of SIRT, alone or in combination with systemic therapy, over the standard-of-care therapy Sorafenib [18,19,20]. However, the retrospective, non-inferiority analysis of these large trials (NEMESIS trial) showed that SIRT offered similar survival rates to Sorafenib, with reduced toxicity and improved quality of life [21].

Moreover, the design of these trials has been criticized [22], while deeper knowledge was gained on SIRT indications and techniques. Today, it is generally recognized that macrovascular invasion involving the main portal trunk should be an exclusion criterion for SIRT [23,24], and that baseline liver function strongly affects treatment outcomes [23,25]. Moreover, we have learned that tumor targeting and personalized dosimetry strongly impact the clinical outcomes of SIRT. The recent prospective Dosisphere trial set a new standard for dosimetry in intermediate/advanced HCC using Y-90 glass microspheres [26] and re-opened the debate on the role of SIRT in intermediate/advanced HCC [14].

Regarding intermediate-stage HCC, little data are available comparing SIRT to TACE, and small randomized studies have not demonstrated the superiority of one treatment over the other [27], although the number of treatments needed to achieve similar tumor response may be lower for SIRT compared to TACE.

Nonetheless, prospective studies and propensity score-matched analyses have shown that, compared to TACE, SIRT is better tolerated and can be associated with higher rates of complete response and downstaging to transplantation and longer time to progression [9,28,29]. Thus, SIRT is becoming appealing in potentially surgical patients or in intermediate-stage patients with tumors involving multiple liver segments.

Over the past few years, SIRT has also emerged as a potentially curative therapy in early-stage HCC. A recent multicenter prospective study reported an 88% objective response rate, with long-lasting tumor response in 62% of patients and 86.6% of patients with a 3-year survival rate, which is comparable to other curative treatments [30]. In the present series, non-early-stage HCCs were treated with SIRT between March and July 2020, probably as a result of the above-mentioned limitations for patients at lower risk of rapid progression.

#### 4.1.2. Intrahepatic Cholangiocarcinoma

Our series included 13.9% inoperable iCC patients treated with SIRT in 2020. Multiple retrospective and prospective series reported good tolerability and objective response rates, with 12–14 months median OS [31,32], comparable to the data reported after TACE [33]. The role of SIRT in iCC is yet not well-defined [7]. Indeed, our series included patients with stable disease after first-line chemotherapy, as well as patients progressing after resection and chemotherapy. SIRT was also performed as first-line treatment in two iCC patients with contraindications to systemic therapies.

Of interest, in a prospective phase 2 study, Edeline et al. showed a longer OS (median 22 months) in selected iCC patients treated with a combination of standard chemotherapy and SIRT in first-line therapy [34]. The longer OS was mainly because 22% of patients were downstaged to resection after this combination therapy, with an 88.9% survival rate 2 years after surgery. The majority of downstaged patients had liver-only disease, with unifocal lesion confined to one hemiliver and no cirrhosis. Thus, in this highly selected “potentially resectable” population, an intensive treatment regimen combing chemotherapy and SIRT could help reducing the tumor load while inducing contralateral liver lobe hypertrophy and ultimately allowing for safe R0 resection.

#### 4.1.3. Liver Metastases

Numerous radiosensitive metastatic lesions may be treated with SIRT, alone or in combination with personalized therapies [35].

In the present series, most of our metastatic patients were affected by mCRC and were treated after the failure of second- or third-line systemic therapy. After the failure of large, randomized studies investigating the role of SIRT as first-line therapy [36], European guidelines consider SIRT as a valid treatment option in unresectable patients with liver-only or liver-dominant oligometastatic mCRC, failing all available systemic treatment options [8]. However, in clinical practice, SIRT is increasingly used also in earlier stages, as consolidation after first- or second-line therapies or to allow for some time of chemo-holiday [11]. The concept of “chemo-holiday” is of particular interest in the context of this pandemic, for patients with preserved liver function, in need for some wash out from drug toxicity, for whom SIRT is able to control the tumor progression with good tolerability, provided adequate tumor targeting and dosimetry.

In fact, for HCC and in mCRC, the tumor response is strictly related to the tumor-absorbed dose, while the liver toxicity should be limited by preliminary assessment of the healthy liver-absorbed dose, which should not exceed certain thresholds [37].

### 4.2. Logistic Challenges

Compared to other LDTs, SIRT represents a more complex procedure, requiring at least two visits, a solid expertise and a strong interdisciplinary collaboration, which involves referring physicians, interventional radiologists, nuclear medicine specialists and medical physicists. This complexity may be even more relevant in the context of the COVID-19 pandemic.

#### 4.2.1. Interdisciplinary Communication

With the lockdown, tumor boards were converted into virtual meetings. This allowed for a better organization of the meeting, since the cases to be presented were prepared and, when possible, shared in advanced, for instance uploading into the system examinations performed in other institutions prior to the meeting.

Our university hospital is divided into two different facilities approximately 4 km apart. The virtual boards eliminated the logistic problems related to this distance, facilitating the attendance to all the involved specialists, in particular the colleagues from the Nuclear Medicine Department. The more regular involvement of the nuclear medicine specialists could have had an impact in the observed increase in SIRT treatments.

Due to these beneficial effects, our tumor boards were maintained virtually after the end of the lockdown.

#### 4.2.2. Hospitalization and Resource Optimization

Prior to definitive treatment, patients are evaluated using some of the following techniques: angiographic mapping, arterial embolization, injection of a radiotracer (or scout dose with Holmium-166 labeled particles) and assessment of its distribution with SPECT/SPECT-CT. This preliminary evaluation is needed to identify possible contraindications to SIRT (such as extrahepatic uptake, high lung shunt fraction and poor tumor targeting) and to calculate the activity to be delivered to tumor. Then, the calculated dose is ordered specifically for each patient on a specific date, and its administration is typically performed after 1–3 weeks. This usually implies two separate hospitalizations; for instance, in our institution, at least 2 days of hospitalization are required for each procedure, mainly because of safety concerns and reimbursement issues.

International surveys have described the impact of COVID-19 in nuclear medicine activity, reporting decreases in diagnostic and therapeutic procedures, and in some countries, insufficient supplies of essential materials, including Tc-99m [38,39,40]. Although COVID-19 also affected this activity in our institution, the Nuclear Medicine Department was maintained as a COVID-19-free ward, and we were able to increase the number of SIRT procedures by reserving two beds each week specifically for both the diagnostic work up and the SIRT treatment, identifying specific days of the week (Tuesday and Thursday, respectively) in order to maintain an efficient turnover.

The pandemic has stimulated an analysis on resource consumption and how to optimize it, starting from the duration of hospitalization. In some countries, SIRT is safely performed as an outpatient procedure [41,42], thanks to the utilization of the trans-radial arterial access that allows for faster patient mobilization [43,44]; the pandemic represents the opportunity to discuss national and local policies and push forward the idea of “ambulatory” SIRT.

When using resin microspheres, same-day SIRT may represent another option to reduce the need for hospitalization, in well-selected patients [45]. Recent papers have also suggested that the preliminary diagnostic work up could be avoided in specific situations, such as patients with small HCC and without TIPS, who are candidates for radiation segmentectomy, since in these conditions, the risk of clinically significant lung shunting is minimal, tumor targeting and extrahepatic uptake can be ruled out using intraprocedural cone-beam CT and dosimetry can be calculated on the basis of the whole liver and target liver volumes on baseline cross-sectional imaging and intraprocedural cone-beam CT [46,47].

In the future, this treatment could be further simplified by introducing “off-the-shelf” vials. The labeled beads have a specific daily decay and can be potentially used any day of the week, once the desired activity to be administered is obtained. SIRT could then become a more expedited treatment, similar to other transarterial therapies. However, to limit the expenses, “off-the-shelf SIRT” implies a relatively high volume of SIRT treatments and rapid turnover.

## 5. Conclusions

In our experience, the first wave of the COVID-19 pandemic witnessed an increased demand for SIRT. In a scenario characterized by delays in oncologic referrals and surgical procedures, limited access to care and difficulties in monitoring side effects of systemic therapies, SIRT represents a safe, manageable and effective treatment to control tumor progression in both primary and secondary liver lesions.

Efficacy and manageability of SIRT will further improve by simplifying the treatment workflow, developing user-friendly and reliable tools for personalized dosimetry and improving interdisciplinary communication.

## Figures and Tables

**Figure 1 jcm-10-04315-f001:**
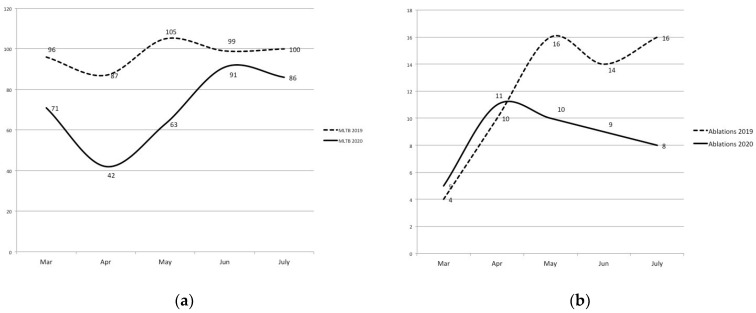
Liver oncologic activity between March and July 2019 and 2020: number of patients discussed at MLTB (**a**), ablations (**b**), transarterial chemoembolizations (**c**) and SIRT (**d**).

**Table 1 jcm-10-04315-t001:** Data of treated patients who underwent SIRT in March–July period: comparison between 2019 and 2020.

		2019	2020	*p*
Number of patients		22	36	
Gender	Male	16 (72.7)	27 (75.0)	0.85
Age (years)	Mean ± SD	68.8 ± 10.6	67.1 ± 11.1	0.55
Type of tumor	Primary	14 (63.6)	27 (75.0)	0.36
	Metastatic	8 (36.4)	9 (25.0)	
Histotype	HCC	12 (54.5)	22 (61.1)	0.51
	ICC	2 (9.1)	5 (13.9)	
	mCRC	6 (27.3)	6 (16.7)	
	Other	2 (9.1) *	3 (8.3) **	
BCLC staging (HCC)	A	3 (25)	0 (0)	0.04
	B	6 (50)	14 (63.6)	
	C	3 (25)	8 (36.4)	
TNM staging (iCC)	III	0 (0)	4 (80)	0.03
	IVa	2 (100)	0 (0)	
	IVb	0 (0)	1 (20)	
Treatment line (metastases)	First	2 (25)	0 (0)	0.23
	Second	3 (37.5)	3 (33.3)	
	Third	3 (37.5)	6 (66.7)	
Type of procedure	Segmental	2 (9.1)	4 (11.1)	0.78
	Unilobar	16 (72.7)	23 (63.9)	
	Bilobar	4 (18.2)	9 (25)	
Type of spheres	Yttrium-90 resin	18 (81.8)	26 (72.2)	0.57
	Yttrium-90 glass	2 (9.1)	3 (8.3)	
	Holmium-166	2 (9.1)	7 (19.5)	

When not otherwise specified, data are given as numbers (and percentages). * Includes: neuroendocrine tumor (*n* = 1) and medullary thyroid carcinoma (*n* = 1). ** Includes: renal cell carcinoma (*n* = 1), uveal melanoma (*n* = 1) and lung cancer (*n* = 1).

**Table 2 jcm-10-04315-t002:** Best radiological response according to tumor type in patients treated in 2020.

Tumor Type	Patients	CR	PR	SD	PD
HCC (*n* = 22)	18	4 (22.2)	9 (50)	3 (16.7)	2 (11.1)
ICC (*n* = 5)	5	1 (20)	0	4 (80)	0
mCRC (*n* = 6)	5	1 (20)	0	2 (40)	2 (40)
Other * (*n* = 3)	2	1 (50)	1 (50)	0	0
Overall (*n* = 36)	30	7 (23.3)	10 (33.3)	9 (30)	4 (13.3)

When not otherwise specified, data are given as numbers (and percentages). * Includes: renal cell carcinoma (*n* = 1), melanoma (*n* = 1) and lung cancer (*n* = 1).

## Data Availability

The data presented in this study are available on request from the corresponding author.

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
