# Peer review of "Management of Liver Tumors during the COVID-19 Pandemic: The Added Value of Selective Internal Radiation Therapy (SIRT)"

_jcm, 2021, doi:10.3390/jcm10194315_

Round 1

Reviewer 1 Report

This article reports a single institution experience with liver-directed therapies during the height of the COVID19 pandemic. Despite the fact that so many medical procedures had been forced to a stop, the number of TACE remained stable while the number of SIRT increased by 64%.

This is a very timely and interesting topic. However, the content of the discussion never really related to the title Management of liver tumors during the COVID-19 pandemic and is a general discussion about SIRT for various tumors. This would be better suited for a SIRT-related review article. The authors mention in 4.2 Logistical challenges that SIRT is a complex procedure involving many disciplines which may be relevant in times of COVID19, but did not further elucidate. What was the general impact of the pandemic on liver directed therapies? The reader would want to know why the authors think there has been an increase in the numbers of SIRT. Is it because surgery and chemotherapies were not available? Would that potentially have an impact on when SIRT should be placed in the treatment line of a patient? Did the pandemic lead to any changes in how SIRT was performed at your institution? Were tumor boards and patient visits in person or virtual? If virtual, was there a better attendance to tumor boards? How were virtual visits perceived by the patient? Virtual tumor boards may have a high attendance and a better efficiency. Would the pandemic be the time to push for change to a one-day protocol where the patient is discharged 4h after SIRT so that hospitalization is avoided? I am not referring to SIRT evaluation and treatment in the same day. I mean separate visits for either, but the hospitalization after intervention could be avoided as it is not necessary from a medical perspective. Of course, if complications occur, a bed should be reserved in advance. (Related to that: Were the patients also hospitalized after pre-treatment evaluation?) As mentioned, reimbursement and country-related specifics of course plays a huge role, but the pandemic could lead or push forward to a change in the policy of hospitalization after SIRT. All this would be great to discuss.

In the last paragraph of the discussion, future outlook (301), the authors mention “off-the-shelf” pre-calibrated vials which in my opinion collides with the later in the conclusion mentioned personalized dosimetry.

Specific remarks:

Use consistent abbreviations: liver-directed therapies was abbreviated LTDs (14, 20) in the abstract and in the manuscript (46) instead of LDTs.

35: Please add disease: coronavirus disease 2019 = COVID-19

Please review typos (for instance: 46: trials, 74: histology, 269: context, 306) and general grammar (for instance: 145: did not differ).

Author Response

This article reports a single institution experience with liver-directed therapies during the height of the COVID19 pandemic. Despite the fact that so many medical procedures had been forced to a stop, the number of TACE remained stable while the number of SIRT increased by 64%.

This is a very timely and interesting topic. However, the content of the discussion never really related to the title Management of liver tumors during the COVID-19 pandemic and is a general discussion about SIRT for various tumors. This would be better suited for a SIRT-related review article.

The authors mention in 4.2 Logistical challenges that SIRT is a complex procedure involving many disciplines which may be relevant in times of COVID19, but did not further elucidate.

Thank you for the valuable inputs provided in your review. The discussion has been modified following your suggestions. In particular:

1) we shortened a little the section regarding the indications for SIRT, trying to underline what was our use of SIRT during and immediately after the lockdown and giving potential explanations of the differences in managing liver tumors, such as:

- treatments in early-stages HCC were mostly postponed, while intermediate-stages unfit for TACE and locally advanced HCCs were preferentially treated with SIRT due to the reduced access to clinical trials investigating new systemic therapies and the willingness of limiting the use of therapies with many side effects (such as sorafenib)

- it is more difficult to specify the reasons to use SIRT in mCRC and iCC since the indications were different for each patient; however, we underlined the fact that many mCRC patients were treated with SIRT to allow some time of suspension from systemic therapies (chemo-holiday)

2) the section regarding the logistic challenges was modified by focusing more on interdisciplinary communications (in particular, the role of virtual tumor boards as suggested) and describing more in detail our local organization and the issues related to hospitalization and resource consumption.

What was the general impact of the pandemic on liver directed therapies? The reader would want to know why the authors think there has been an increase in the numbers of SIRT. Is it because surgery and chemotherapies were not available? Would that potentially have an impact on when SIRT should be placed in the treatment line of a patient?

As mentioned above, the use of SIRT mostly replaced systemic therapies particularly for HCC and mCRC. However, the data we are presenting cannot be used as a means to suggest a different place of SIRT in treatment algorithms, although we present what we believe could be some relevant literature for re-assessing the role of SIRT in different clinical scenarios.

Did the pandemic lead to any changes in how SIRT was performed at your institution? Were tumor boards and patient visits in person or virtual? If virtual, was there a better attendance to tumor boards? How were virtual visits perceived by the patient? Virtual tumor boards may have a high attendance and a better efficiency.

We added a paragraph specifically addressing the role of virtual tumor boards and what they added in our daily practice.

Would the pandemic be the time to push for change to a one-day protocol where the patient is discharged 4h after SIRT so that hospitalization is avoided? I am not referring to SIRT evaluation and treatment in the same day. I mean separate visits for either, but the hospitalization after intervention could be avoided as it is not necessary from a medical perspective. Of course, if complications occur, a bed should be reserved in advance. (Related to that: Were the patients also hospitalized after pre-treatment evaluation?) As mentioned, reimbursement and country-related specifics of course plays a huge role, but the pandemic could lead or push forward to a change in the policy of hospitalization after SIRT. All this would be great to discuss.

We added a sentence as suggested by the reviewer “Despite this organization, the pandemic represented a stimulus for analyzing the possibility of resources’ optimization, and consider changes in national and local policies for outpatient procedures,”

Specifically

In the last paragraph of the discussion, future outlook (301), the authors mention “off-the-shelf” pre-calibrated vials which in my opinion collides with the later in the conclusion mentioned personalized dosimetry.

We modified the sentence in the attempt to better explain the concept of “off-the-shelf” vials; basically, the vial can be ordered in advanced (with higher calibration) and kept on the shelf until adequate activity is reached. This is more simple for resin spheres since they can be now ordered with different pre-calibrations, but it could potentially work also for holmium and glass spheres.

Specific remarks:

Use consistent abbreviations: liver-directed therapies was abbreviated LTDs (14, 20) in the abstract and in the manuscript (46) instead of LDTs.

We corrected the abbreviations

35: Please add disease: coronavirus disease 2019 = COVID-19

“coronavirus disease 2019” was added when first mentioning the abbreviation

Please review typos (for instance: 46: trials, 74: histology, 269: context, 306) and general grammar (for instance: 145: did not differ).

We verified spelling and grammar; hope the text is more comprehensible and correct now

Reviewer 2 Report

The manuscript regards the management of liver tumors during the very special COVID 19 pandemic period. The authors evaluate the changes in the therapeutic strategies due to the restrictions caused by the lockdown. They demonstrate an increase of patients treated by SIRT during this period and explain why it was preferred to other loco regional therapies. The review is well written and well structured. This is an interesting article demonstrating the advantages of SIRT compared to other similar effectiveness therapies.  

Some aspects need explanations and clarifications:

  • P3 line 111: The lockdown was in Italy between March and May 2020. However, the patients were analysed between march and July; with a large increase of SIRT in July despite the end of the lookdown.
  • P3 line 114 and P6 line 168: You demonstrate a large decrease of percutaneous ablations compared to the previous year during the pandemic period and an increase of SIRT treatments especially for HCC patients with large or multiples tumors. However, percutaneous ablations are only indicated for small liver metastases or HCC at an early stage. Could you explain this difference? In your opinion, percutaneous ablations were replaced by SIRT?
  • Table 1 P4 line 140: “mean tumor absorbed dose”. Patients were treated by resin, glass and Holmium microspheres. Each type of treatment is defined by a proper dose response correlation (twofold effectiveness between glass and resin). Dosimetry has a minimal interest in this paper, please detail or modify. Moreover, patients were treated by resin, glass and Holmium microspheres. It could be interesting to precise the differences between each others and in which conditions you used one type compared to others.  
  • P6 line 190: “officially” is not adequate. Moreover, Recent ESMO guidelines consider SIRT in the strategy.
  • P6 line 217: It is also important to precise and highlight the differences between TACE and SIRT (number of sessions…).
  • P 7 line 258: Not totally true…. See the HEPAR PLUS trial using QuiremSpheres.
  • P7 line 262: “tumor extent…..not exceed 50% of the tumor volume”. Precise the risk of REILD in these conditions. Despite a large tumor extension, it is also the possible to limit the toxicity using optimized activities planned on the non-tumoral liver.
  • P7 line 267: Please precise also that SIRT is very technical and requires a solid expertise.
  • P7 line 271 “radiotraceur” could be more detailed. Morevorer, not true for Quirem Spheres.
  • P8 line 297: Many experts and recommendations do not recommend this strategy…99mTc- MAA is very useful, mainly to avoid extrahepatic uptakes and cannot be replaced actually by cone-beam CT. Moreover, the study of Gabr et al defined only segmental treatments with little tumors.
  • In the last part of your manuscript, you could also discuss about QuiremScout dose using Holmium spheres.

Details:

  • Table 1 P4 line 140: “SIRT treatment extension”. Please change by “type of procedures” or other.
  • P5 line 154: please specify the mean follow up for the radiological outcome.
  • Table 2 P5 line 160: The column “ no imaging” is not necessary. You could precise it in material and methods.
  • P6 lines 202-203: “in patients with microvascular invasion…..”This 

    The manuscript regards the management of liver tumors during the very special COVID 19 pandemic period. The authors evaluate the changes in the therapeutic strategies due to the restrictions caused by the lockdown. They demonstrate an increase of patients treated by SIRT during this period and explain why it was preferred to other loco regional therapies. The review is well written and well structured. This is an interesting article demonstrating the advantages of SIRT compared to other similar effectiveness therapies.  

    Some aspects need explanations and clarifications:

    • P3 line 111: The lockdown was in Italy between March and May 2020. However, the patients were analysed between march and July; with a large increase of SIRT in July despite the end of the lookdown.
    • P3 line 114 and P6 line 168: You demonstrate a large decrease of percutaneous ablations compared to the previous year during the pandemic period and an increase of SIRT treatments especially for HCC patients with large or multiples tumors. However, percutaneous ablations are only indicated for small liver metastases or HCC at an early stage. Could you explain this difference? In your opinion, percutaneous ablations were replaced by SIRT?
    • Table 1 P4 line 140: “mean tumor absorbed dose”. Patients were treated by resin, glass and Holmium microspheres. Each type of treatment is defined by a proper dose response correlation (twofold effectiveness between glass and resin). Dosimetry has a minimal interest in this paper, please detail or modify. Moreover, patients were treated by resin, glass and Holmium microspheres. It could be interesting to precise the differences between each others and in which conditions you used one type compared to others.  
    • P6 line 190: “officially” is not adequate. Moreover, Recent ESMO guidelines consider SIRT in the strategy.
    • P6 line 217: It is also important to precise and highlight the differences between TACE and SIRT (number of sessions…).
    • P 7 line 258: Not totally true…. See the HEPAR PLUS trial using QuiremSpheres.
    • P7 line 262: “tumor extent…..not exceed 50% of the tumor volume”. Precise the risk of REILD in these conditions. Despite a large tumor extension, it is also the possible to limit the toxicity using optimized activities planned on the non-tumoral liver.
    • P7 line 267: Please precise also that SIRT is very technical and requires a solid expertise.
    • P7 line 271 “radiotraceur” could be more detailed. Morevorer, not true for Quirem Spheres.
    • P8 line 297: Many experts and recommendations do not recommend this strategy…99mTc- MAA is very useful, mainly to avoid extrahepatic uptakes and cannot be replaced actually by cone-beam CT. Moreover, the study of Gabr et al defined only segmental treatments with little tumors.
    • In the last part of your manuscript, you could also discuss about QuiremScout dose using Holmium spheres.

    Details:

    • Table 1 P4 line 140: “SIRT treatment extension”. Please change by “type of procedures” or other.
    • P5 line 154: please specify the mean follow up for the radiological outcome.
    • Table 2 P5 line 160: The column “ no imaging” is not necessary. You could precise it in material and methods.
    • P6 lines 202-203: “in patients with microvascular invasion…..”This sentence is unclear (precise the effectiveness of the therapy).
    • P7 line 242: “Hemi liver” or lobe?
    • P8 line 279: The two days hospitalization is mainly performed to control the vascular access of the catheter.
    • P8 line 295: For resin microspheres.
    • P 8 line 306: “ian” => an
    sentence is unclear (precise the effectiveness of the therapy).
  • P7 line 242: “Hemi liver” or lobe?
  • P8 line 279: The two days hospitalization is mainly performed to control the vascular access of the catheter.
  • P8 line 295: For resin microspheres.
  • P 8 line 306: “ian” => an

Author Response

The manuscript regards the management of liver tumors during the very special COVID 19 pandemic period. The authors evaluate the changes in the therapeutic strategies due to the restrictions caused by the lockdown. They demonstrate an increase of patients treated by SIRT during this period and explain why it was preferred to other loco regional therapies. The review is well written and well structured. This is an interesting article demonstrating the advantages of SIRT compared to other similar effectiveness therapies.  

Some aspects need explanations and clarifications:

  • P3 line 111: The lockdown was in Italy between March and May 2020. However, the patients were analysed between march and July; with a large increase of SIRT in July despite the end of the lookdown.

As now mentioned, with added June and July to account for the effects of the lockdown considering that the phase of exiting was at least as complex as the lockdown itself. The increased number of SIRT in July reflects the increased availability of beds in other departments in June-July and the need to deplete the waiting list before the summer break in August (data not reported in the text)

  • P3 line 114 and P6 line 168: You demonstrate a large decrease of percutaneous ablations compared to the previous year during the pandemic period and an increase of SIRT treatments especially for HCC patients with large or multiples tumors. However, percutaneous ablations are only indicated for small liver metastases or HCC at an early stage. Could you explain this difference? In your opinion, percutaneous ablations were replaced by SIRT?

As now mentioned in the discussion, ablations were mostly postponed; SIRT did not replace ablations, but we believe that the increased number of SIRT procedures was the result of the reduced utilization of systemic therapies, particularly in HCC patients.

  • Table 1 P4 line 140: “mean tumor absorbed dose”. Patients were treated by resin, glass and Holmium microspheres. Each type of treatment is defined by a proper dose response correlation (twofold effectiveness between glass and resin). Dosimetry has a minimal interest in this paper, please detail or modify. Moreover, patients were treated by resin, glass and Holmium microspheres. It could be interesting to precise the differences between each others and in which conditions you used one type compared to others.  

As suggested, the column of mean absorbed dose was deleted, since it has a minimal interest in this paper. Accordingly, we believe that describing when and how we use the different types of particles is beyond the scope of this manuscript.

  • P6 line 190: “officially” is not adequate. Moreover, Recent ESMO guidelines consider SIRT in the strategy.

The sentence has been deleted because we modified the entire discussion according to the suggestion of the other reviewer.

  • P6 line 217: It is also important to precise and highlight the differences between TACE and SIRT (number of sessions…).

We added the sentence related to the reduced number of treatments required for SIRT compared to TACE; however, we did not add much more information on this specific topic to maintain the flow of the discussion

  • P 7 line 258: Not totally true…. See the HEPAR PLUS trial using QuiremSpheres.

The sentence was modified; the reference of the Hepar Plus study was added

  • P7 line 262: “tumor extent…..not exceed 50% of the tumor volume”. Precise the risk of REILD in these conditions. Despite a large tumor extension, it is also the possible to limit the toxicity using optimized activities planned on the non-tumoral liver.

The sentence was modified, mentioning a recent paper investigating dose-toxicity and dose-efficacy in mCRC

  • P7 line 267: Please precise also that SIRT is very technical and requires a solid expertise.

The concept of solid expertise was added

  • P7 line 271 “radiotraceur” could be more detailed. Morevorer, not true for Quirem Spheres.

We did not add many more details, since it is not the objective of the study to describe precisely how to perform SIRT; however we distinguished between radiotracer and Quirem scout dose.

  • P8 line 297: Many experts and recommendations do not recommend this strategy…99mTc- MAA is very useful, mainly to avoid extrahepatic uptakes and cannot be replaced actually by cone-beam CT. Moreover, the study of Gabr et al defined only segmental treatments with little tumors.

We specified that this is for radiation segmentectomy

  • In the last part of your manuscript, you could also discuss about QuiremScout dose using Holmium spheres.

We did not add any specific comments on Scout dose since, as already mentioned, it is beyond the objective of the study

Details:

  • Table 1 P4 line 140: “SIRT treatment extension”. Please change by “type of procedures” or other.

We modified as suggested

  • P5 line 154: please specify the mean follow up for the radiological outcome.

A sentence was added

  • Table 2 P5 line 160: The column “ no imaging” is not necessary. You could precise it in material and methods.

We deleted as requested

  • P6 lines 202-203: “in patients with microvascular invasion…..”This sentence is unclear (precise the effectiveness of the therapy).

This comment is not clear; however, the sentence has been modified during the editing of the discussion

  • P7 line 242: “Hemi liver” or lobe?

We kept the concept of “hemiliver” as reported by the authors of the paper

  • P8 line 279: The two days hospitalization is mainly performed to control the vascular access of the catheter.

I guess it depends on local regulations; we use closure devices for the vascular access that do not need two days of monitoring. In our hospital two days of hospitalization are required mainly for radiation protection issues (according to local regulations) and reimbursement policies (the procedure would not receive sufficient reimbursement if performed as day case or with only one day of hospitalization)

  • P8 line 295: For resin microspheres.

Added, as requested

  • P 8 line 306: “ian” => an

Corrected, as pointed out